# Utilizing Tympanic Membrane Temperature for Earphone-Based Emotion Recognition

**DOI:** 10.3390/s25144411

**Published:** 2025-07-15

**Authors:** Kaita Furukawa, Xinyu Shui, Ming Li, Dan Zhang

**Affiliations:** 1Department of Psychological and Cognitive Sciences, Tsinghua University, Beijing 100084, China; gcht21@tsinghua.org.cn (K.F.); shuixy19@mails.tsinghua.edu.cn (X.S.); m-li20@mails.tsinghua.edu.cn (M.L.); 2Tsinghua Laboratory of Brain and Intelligence, Beijing 100084, China

**Keywords:** affective computing, physiological asymmetry, tympanic membrane temperature, earphones, wearable devices

## Abstract

Emotion recognition by wearable devices is essential for advancing emotion-aware human–computer interaction in real life. Earphones have the potential to naturally capture brain activity and its lateralization, which is associated with emotion. In this study, we newly introduced tympanic membrane temperature (TMT), previously used as an index of lateralized brain activation, for earphone-based emotion recognition. We developed custom earphones to measure bilateral TMT and conducted two experiments consisting of emotion induction by autobiographical recall and scenario imagination. Using features derived from the right–left TMT difference, we trained classifiers for both four-class discrete emotion and valence (positive vs. negative) classification tasks. The classifiers achieved 36.2% and 42.5% accuracy for four-class classification and 72.5% and 68.8% accuracy for binary classification, respectively, in the two experiments, confirmed by leave-one-participant-out cross-validation. Notably, consistent improvement in accuracy was specific to models utilizing right–left TMT and not observed in models utilizing the right–left wrist skin temperature. These findings suggest that lateralization in TMT provides unique information about emotional state, making it valuable for emotion recognition. With the ease of measurement by earphones, TMT has significant potential for real-world application of emotion recognition.

## 1. Introduction

As interaction with intelligent machines becomes more common, their ability to recognize human emotions is essential for enriching our daily experiences. By integrating emotional intelligence into machine systems, they can tailor their responses to suit our feelings. Researchers in affective computing have been trying to enable machines to recognize, express, and “have” emotion [1], laying the foundation for emotion-aware human–computer interaction. Emotion recognition is the first step in affective computing, and researchers are exploring various modalities, including visual, auditory, textual, and physiological, to estimate human emotion [2]. Among these modalities, physiological signals offer a direct window into a person’s internal state. Unlike facial expressions, voice, or text, which can be consciously controlled or masked, physiological signals tend to be involuntary and thus more objective indicators of emotion [3]. Emotion recognition based on physiological signals remains in the early stages of development, with most research conducted under controlled laboratory conditions [4]. In these settings, researchers use specialized equipment to collect data such as electroencephalography (EEG) via multi-electrode headsets, or electrocardiography (ECG) through electrodes placed on the body [5]. These methods are often expensive, require extensive setup, and are not well-suited for everyday use. For the real-world applications of emotion recognition, there is a growing need for ubiquitous and non-invasive methods of collecting physiological signals.

Wearable devices meet these requirements, as they are widely adopted and naturally integrated into daily life. Recent advancements in sensor technology have enabled wearable devices to capture a wide range of physiological signals, and previous studies have utilized heart rate, heart rate variability, ECG, skin temperature, respiration, EMG, galvanic skin response (GSR), photoplethysmography (PPG), and pulse rate variability for emotion recognition [6,7,8]. However, most studies have focused on physiological signals associated with the peripheral nervous system [8], and the studies targeting the central nervous system, especially the brain, remain limited. In the recent theory of emotion, the brain is the place where emotions are constructed [9,10], and it is one of the most important areas to explore for the relationship with emotion. Therefore, wearable devices capable of capturing brain-related signals are essential for further advancing the research field.

Earphones emerge as a platform for capturing brain activity, given their anatomical and functional advantages [11]. The ear presents a promising site for physiological sensing due to its proximity to the brain and its rich vascular structure, which supports non-invasive access to brain-related signals [12]. In addition, earphones are among the most widely used wearable devices in daily life. Leveraging these advantages, several studies have demonstrated the feasibility of collecting EEG using earphone-based sensors for emotion recognition [13,14]. For instance, Athavipach et al. [13] developed an in-ear EEG measurement device with a dry electrode on an earpiece. They collected EEG signals during an emotion elicitation experiment using picture stimuli and achieved classification accuracies of 73.01% for valence, 75.70% for arousal, and 59.23% for four-category emotion classification. Similarly, Mai et al. [14] designed an ear EEG device with electrodes positioned behind the ear. Using data collected during the video watching experiment, they achieved 94.87% accuracy for user-dependent and 84.68% for user-independent valence classification. These findings illustrate the potential of earphone-based sensing as a practical approach for emotion recognition in real-world settings.

The current study introduces a new modality for earphone-based emotion recognition: tympanic membrane temperature (TMT). TMT is commonly used as an index of body temperature [15], and it was also used to estimate brain temperature given its proximity to the hypothalamus and internal carotid artery, supplying blood to the brain [16,17]. In psychological studies, researchers used TMT as an indicator of brain lateralization [18,19,20], and several studies reported the association between bilateral asymmetry in TMT and emotion [21,22,23]. While TMT shares similarities with other physiological signals previously used in emotion recognition, it also possesses distinct characteristics. Like skin temperature, it measures thermal activity; however, while skin temperature reflects the peripheral nervous system, TMT is considered to reflect the central nervous system. EEG also targets brain-related activity, but it uses different measurements, and the location of the tympanic membrane is deeper inside the head in contrast to EEG electrodes placed on the surface of the scalp. These differences indicate that TMT may offer unique insights into emotional states not accessible through other modalities. Moreover, TMT has the advantage of accessibility. TMT measurement is non-invasive, and the temperature can be obtained instantly by an infrared sensor inside the ear canal. The sensor is small enough to be embedded inside wearable devices, and researchers demonstrated TMT could be measured by earphones [24,25]. Earphones are naturally worn on both sides of a head and are suitable devices to capture the lateralization of TMT. However, the earphone-based TMT measurement has never been used for emotion recognition, and previous psychological studies have typically measured TMT only a few times during an experiment using shotgun-shaped devices [21,22,23]. Consequently, the temporal dynamics of TMT during emotional experiences and its utility for emotion recognition remain largely unexplored.

To the best of our knowledge, this is the first study to use TMT for emotion recognition. In the absence of commercially available devices that can continuously measure TMT, we first designed and developed custom earphone-type TMT thermometers. Using the custom earphones, we conducted two experiments utilizing different emotion induction methods, autobiographical recall and scenario imagination, to offer a holistic evaluation of TMT for emotion recognition. The continuous measurement of TMT enabled us to investigate the signal in detail, and we found that TMT responds differently to distinct emotional states and can be used for emotion recognition. The primary contributions of this study are as follows:Introducing TMT as a novel physiological signal for emotion recognition;Developing custom earphone-based devices for naturalistic, continuous TMT measurement;Offering a comprehensive assessment of TMT for emotion recognition by utilizing autobiographical recall and scenario imagination methods;Demonstrating that right-to-left difference in TMT can effectively support emotion classification across different experiments.

## 2. Related Work

### 2.1. The Association Between TMT Lateralization and Emotion

Previous studies that explore the relationship between TMT and emotion have drawn on the lateralized brain theory of emotion, where researchers measure lateralization in TMT to detect lateralization in brain activity [18,19,20]. The lateralized brain theory typically associates the left hemisphere with positive or approaching emotion and the right hemisphere with negative or withdrawing emotion [26]. This hypothesis is supported by a wide range of studies, including behavioral research on patients with brain lesions [27], EEG studies [28,29,30], and neuroimaging studies [31,32,33].

Building on this foundation, several studies have attempted to examine lateralized brain activity during emotional experiences through lateralization in TMT. Researchers have investigated TMT because it is simple, non-invasive, and significantly less expensive than alternative methods [21,22]. For instance, Helton et al. [21] found a trend that right TMT increased more than left TMT after exposure to negative picture stimuli compared with neutral picture stimuli. Propper et al. [22] also evaluated individual differences in TMT and found a positive correlation between the absolute value of the right-to-left difference in TMT (right–left TMT) and anger or hostility scores on a mood state questionnaire. In subsequent study, Propper et al. [23] found a positive correlation between the absolute value of right–left TMT and anger induced by music stimuli. These findings suggest that TMT lateralization may be associated with emotional states, yet consistent patterns have not been found. One limitation of prior research is the low sampling frequency, as TMT was typically measured only a few times during the experiment.

The current study addresses this limitation by introducing continuous TMT measurement and applying extensive statistical analyses to explore the relationship in depth. This approach allows for a detailed examination of how TMT changes with emotional experiences, advancing our understanding of TMT as a physiological marker of emotion.

### 2.2. Measurement of TMT

Tympanic membrane temperature (TMT) has been measured using several methods, including medical probes, handheld infrared thermometers, and earphone-type infrared sensors. In early studies, TMT was measured using a medical probe that was placed directly on the tympanic membrane during experiments [34,35]. While this method offers high precision, it requires professional handling to avoid damaging the membrane, making it unsuitable for general use.

The most common method for TMT measurement today involves handheld shotgun-type infrared thermometers, such as the Braun ThermoScan^®^. This approach allows researchers to obtain TMT by placing the thermometer nozzle into the participant’s ear canal for a few seconds. Although this method is simple, it lacks wearability. All previous studies investigating the relationship between TMT and emotion used this type of thermometer, measuring the right and left TMT sequentially [21,22,23]. Despite its ease, the shape of shotgun-type thermometers makes them unsuitable for continuous measurement of TMT.

To address these limitations, some researchers have explored earphone-type devices for TMT measurement. For example, Kiya et al. [24] developed an earphone-based sensor for monitoring TMT during surgery, and Yamakoshi et al. [25] designed a device for race car drivers during practice sessions. Both studies demonstrated that earphone-based TMT measurement is valid and feasible in applied contexts. Building on these examples, we designed a custom earphone-based device for our study.

## 3. Materials and Methods

### 3.1. Development of Earphone-Type Thermometer

In our custom earphones, an infrared sensor module was embedded in the front of the speaker within an earphone shell (Figure 1a–c). This design allows the user to listen to sound while TMT is being measured. We used an infrared sensor (MLX90614ESF-DCA, Melexis NV, Ypres, Belgium) with medical accuracies for 36 to 38 °C and a resolution of 0.02 °C, which was designed to measure human body temperature. The sensor was factory-calibrated, and the temperature came with digital output via SMBus. The data was sampled at 20 Hz using a microcontroller (SEEEDUINO XIAO, Seeed Technology Co., Ltd., Shenzhen, China) equipped with a low-power ARM Cortex-M0+ processor. The circuit used an I2C communication setup where a microcontroller connected to a computer reads data from two infrared temperature sensors. We also developed wristbands to measure dorsal wrist skin temperature using the same circuit as the earphones (Figure 1d,e). Wrist skin temperature was measured as a baseline for comparison with TMT. We chose wrist skin temperature because it can be measured by a wristband, one of the most common wearable devices in the real world beside earphones. The material cost for one set of earphones or wristbands was approximately USD 55 (infrared sensors: USD 44; a microcontroller: USD 5; other components: USD 6). Details of the development, as well as validation experiment results, are in the Appendix A.

### 3.2. Overview of Experiments

We utilized two common emotion induction methods in psychological studies: autobiographical recall and scenario imagination [36,37]. Autobiographical recall involves participants reflecting on past events that elicited strong emotions, and scenario imagination requires participants to imagine situations that could happen in real life. By employing these methods, our study aimed to provide a comprehensive exploration of TMT for emotion recognition. A summary of the experimental flow is provided in Figure 2.

A total of 10 participants from a drama club at Tsinghua University participated in Experiments 1 and 2. All participants had at least one year of acting experience (five females and five males; age: *M* = 22.5, *SD* = 2.2). Experiment 1 was conducted alongside Experiment 2, and ethical approvals for both experiments were obtained from the local ethical committee at the Department of Psychological and Cognitive Sciences, Tsinghua University (IRB Approval 2023 No. 12). All participants provided informed consent before the experiments.

### 3.3. Experiment 1: Autobiographical Recall Experiment

Experiment 1 consisted of four blocks, each targeting a specific discrete emotion: fear, sadness, joy, or love. The four blocks were randomly ordered for each participant. Each block consisted of two sub-blocks for emotion induction. At the beginning of each block, participants provided short memos of two life events when they experienced the target emotion. Then, one of the two memos was randomly selected for each sub-block to prompt participants to recall the specific event. Each sub-block had two parts. In the first part, participants saw the prompt on a screen and wrote down the details of the event for three minutes; in the second part, they recalled the event for two minutes without writing. After this five-minute session, participants rated eight discrete emotion items (anger, disgust, fear, sadness, amusement, inspiration, joy, and tenderness) on horizontal scales ranging from one to nine. The time length of each task was decided by referencing a previous study that measured TMT before and after a five-minute emotion induction session [23], which was also comparable to the minute-level time scales commonly used in other physiological studies on emotion [38,39,40,41]. The second sub-block began ten seconds after the rating of the first sub-block, and participants rested for at least one minute to return to a neutral emotional state between blocks.

At the beginning of the experiment, participants practiced the experimental task in a neutral emotion block with a sub-block to become familiar with the procedure. All instructions in the experiment were presented in Chinese on a desktop screen using PsychoPy v2022.2.5 software. TMT from both ears and skin temperatures from both wrists were measured for our experiment, and 24-channel EEG data from a wired device were measured for other purposes (not included in the present study). TMT and wrist skin temperature measured during the two-minute recalling task were labeled by the target emotion and categorized as either positive emotions (joy and love) or negative emotions (fear and sadness). As a result, we obtained a total of 80 data samples (120 s length each) for each TMT and wrist skin temperature measurement, derived from two sub-blocks for each of the four emotions from the 10 participants, with discrete emotion (fear, sadness, joy, or love) and valence (positive or negative) labels.

### 3.4. Experiment 2: Scenario Imagination Experiment

Experiment 2 was conducted after Experiment 1, with a minimum of 10 min of rest between the two experiments. During this period, participants were instructed to walk around the experiment room, relax, and freely engage in activities to refresh themselves. Experiment 2 followed a similar design to Experiment 1 but used scenario imagination instead of autobiographical recall. We asked participants to read a scenario for the first three minutes and imagine themselves in the scenario for the next two minutes in each sub-block. Compared with prior studies [42], relatively long scripts (310–785 Chinese characters) were used so that participants could continuously imagine a single scenario for two minutes. The emotion-inducing scenarios were developed by the experimenters with the assistance of Generative AI (GPT-3.5) and reviewed by an experienced scenario writer in the college drama club.

The experiment consisted of four blocks targeting the same four discrete emotions, presented in the same order as in Experiment 1. After each five-minute sub-block, participants rated their feelings during the imagination task using the same scales as in Experiment 1. All instructions and scripts were presented in Chinese on a desktop screen using PsychoPy v2022.2.5 software. The physiological signals were recorded in the same way as Experiment 1, and TMT and wrist skin temperature measured during the two-min imagination task were labeled by the target emotion and categorized into positive or negative emotions. Like Experiment 1, we obtained 80 data samples (120 s length each) for each TMT and wrist temperature measurement from the 10 participants, with both discrete emotion and valence labels.

## 4. Data Analysis

### 4.1. Descriptive Analysis of Emotion Ratings

To evaluate whether the participants in each experiment successfully experienced the target emotions, we performed descriptive statistics of their emotion ratings. Specifically, we calculated the mean and standard deviation of the participants’ self-reported ratings on eight predefined emotion scales, grouped by the target emotion and by valence of the stimuli.

### 4.2. Preprocess of Temperature Data

The collected data of TMT and wrist skin temperature showed a few artifacts characterized by sudden changes in the temperature signals. This noise could be attributed to the movements of the participant, and we took the following noise reduction process before further analysis. First, the original signal (20 Hz) was resampled at 1 Hz by calculating the mean value within each non-overlapping one-second interval. Subsequently, the one-second interval differences in the temperature were calculated for each participant, and any samples with a value higher or lower than five times the standard deviation from the mean of the differences were regarded as an outlier and set to zero. Finally, the signal was constructed using the modified interval differences. The outlier was only found in the data from two to three participants in each experiment, with a maximum of 0.3% of total recording data from each participant.

### 4.3. Analysis of Temporal Mean Temperature

Based on the findings in previous studies [21,22,23], we assumed right–left TMT would change differently across different emotions. To investigate the temporal changes in right–left TMT under different emotional conditions, we employed nonparametric cluster-based statistical testing [43] that was originally developed to evaluate the spatiotemporal difference in EEG or MEG data across experimental conditions. We used the first time point of each data sample as a baseline, subtracting it from all subsequent time points in the sample. For each emotional condition of a participant, we calculated the mean of right–left TMT at each time point. Then, we randomly permuted the emotional condition labels within each participant by shuffling the conditions assigned to each trial. The number of possible permutations for *c* emotional conditions is *c*! for each participant, resulting in (*c*!)*^n^* possible patterns for *n* participants. At each time point, we calculated a *t*-value from a paired *t*-test between the positive and negative emotion conditions, as well as an *F*-value from repeated measures ANOVA across different discrete emotion conditions for within-participant comparisons. Next, we identified time points where the test statistics exceeded the threshold corresponding to a *p*-value of 0.05. Using temporal adjacency, we formed clusters by grouping consecutive time points from the selected test statistics. We took the sum of the test statistics for each cluster as a cluster-level test statistic. We repeated this process for 1000 random permutations, selecting the largest absolute value of cluster-level test statistics from each permutation to form a null distribution. The observed cluster-level statistics were then compared to the null distribution, and (two-tailed) *p*-values were calculated as the proportion of permutations in the null distribution where the absolute value of the permutation’s cluster-level test statistic was larger than or equal to the absolute value of the observed cluster-level test statistics.

We also performed this analysis for right-to-left difference in wrist skin temperature (right–left wrist skin temperature), simply replacing right–left TMT with the right–left wrist skin temperature. This comparison was used to analyze whether the change in right-left TMT is unique or like temperature changes in other parts of the body.

### 4.4. Classification

In this analysis, we conducted emotion recognition using a single modality approach, focusing on right–left TMT, and compared the results with right–left wrist skin temperature. The evaluation was performed using leave-one-participant-out validation, where models trained on data from all participants except one were tested on the excluded participant’s data. Importantly, the same model was tested in this way in two different experiments to assess its generalizability.

To effectively capture the temporal change in the signal, we applied the following feature extraction techniques. First, we sliced each data sample into 20 s time windows with 10 s overwrapping strides. Next, we calculated the mean, minimum, maximum, and Shannon entropy values [44] for each time window. We then concatenated the features from all-time windows to create a one-dimensional feature vector for each data sample. After the feature extraction, we applied min–max normalization for the features within each participant to reduce cross-participant difference by scaling each feature to a range of zero to one.

The classification was conducted at two levels: valence classification (a binary classification of positive vs. negative emotion) and discrete emotion classification (a four-class classification of fear, sadness, joy, or love). We employed three classifiers, all implemented using scikit-learn version 1.5.2, with the following parameters:Gaussian Naïve Bayes (GNB): We used the GaussianNB classifier with default parameters.Support Vector Machine (SVM): We employed an SVC with a radial basis function kernel. The penalty parameter C was set to 0.5, with other parameters set to default.Multilayer Perceptron (MLP): Our MLPClassifier consists of three hidden layers, hidden_layer_sizes = (300, 300, 300), with other parameters set to default.

Classification accuracies from the leave-one-participant-out cross-validation were compared to random chance (1/number of classes) using a one-sample *t*-test. Similarly to the analysis of temporal mean temperature, we also evaluated models using wrist skin temperature instead of TMT. The wrist temperature model was constructed following the same step as the TMT model, replacing the right–left TMT with the right–left wrist skin temperature. We compared the highest mean accuracy among the three classifiers using right–left TMT with that of right–left wrist skin temperature for each classification task.

## 5. Results

### 5.1. Emotion Ratings

In both experiments, joy received the highest ratings for tasks targeting joy (Experiment 1: *M* = 7.3, *SD* = 0.9; Experiment 2: *M* = 7.3, *SD* = 0.9), fear for tasks targeting fear (Experiment 1: *M* = 7.6, *SD* = 1.0; Experiment 2: *M* = 8.1, *SD* = 0.9), and sadness for tasks targeting sadness (Experiment 1: *M* = 7.4, *SD* = 1.0; Experiment 2: *M* = 8.0, *SD* = 1.0). Tenderness was highest for tasks targeting love (Experiment 1: *M* = 6.6, *SD* = 2.4; Experiment 2: *M* = 7.5, *SD* = 1.3). Joy had the highest mean rating for the positive emotion tasks in both experiments (Experiment 1: *M* = 6.6, *SD* = 1.8; Experiment 2: *M* = 7.0, *SD* = 1.2), whereas sadness had the highest for negative emotion tasks (Experiment 1: *M* = 6.2, *SD* = 2.6; Experiment 2: *M* = 6.4, *SD* = 2.4). The observed emotion ratings were consistent with the target emotions, suggesting that participants successfully induced the target emotions during the experiments. The results for the emotion ratings are provided in Table 1.

### 5.2. Temporal Temperature Differences

Figure 3a illustrates the mean of right–left TMT for tasks grouped by positive and negative emotions in each experiment. In Experiment 1, we found clusters with *p*-values less than 0.05 between 13 and 16 s (*p* = 0.05) and between 18 and 30 s (*p* = 0.01). In Experiment 2, a significant cluster was observed between 76 and 83 s (*p* = 0.02). In these periods, the means of right–left TMT for negative emotions were lower than those for positive emotions. Figure 3b shows the mean of right–left wrist skin temperature, grouped by positive and negative emotions. We observed a trend where the mean right–left wrist skin temperature for positive emotions was higher than that for negative emotions during most parts of the tasks. However, the *t*-values did not reach threshold (*t*(9) < 2.26) at any time point in either experiment. Figure 3c presents the mean of right–left TMT grouped by targeted discrete emotions. Although *F*-values exceeded the threshold at multiple time points (*F*(3,27) > 2.96) in both experiments, we did not find any clusters with *p*-values less than 0.05. Figure 3d illustrates the mean of right–left wrist skin temperature by the targeted discrete emotions. The *F*-values remained below the critical threshold (*F*(3,27) < 2.96) at all time points in both experiments, indicating no significant differences.

### 5.3. Classification Results

Table 2 summarizes the classification results from Experiment 1. The classification model using right–left TMT achieved accuracies higher than the chance level (>50%) in positive versus negative (valence) classification for all classifiers: 63.7 ± 13.1% (95% CI: 54.3–73.1%, *t*(9) = 3.16, *p* = 0.01) for GNB, 67.5 ± 13.9% (95% CI: 57.6–77.4%, *t*(9) = 3.77, *p* < 0.01) for SVM, and 72.5 ± 16.6% (95% CI: 60.6–84.4%, *t*(9) = 4.07, *p* < 0.01) for MLP. The right–left TMT model also achieved above-chance-level (>25%) accuracy in discrete emotion classification with accuracies of 36.2 ± 8.8% (95% CI: 29.9–42.5%, *t*(9) = 3.86, *p* < 0.01) for GNB and 35.0 ± 12.2% (95% CI: 26.3–43.7%, *t*(9) = 2.45, *p* = 0.03) for SVM. In contrast, none of the classifiers utilizing right–left wrist skin temperature achieved accuracies higher than the chance level accuracy for either valence or discrete emotion classifications.

Table 3 presents the classification results from Experiment 2. All classifiers utilizing right–left TMT achieved accuracies higher than the chance level (>50%) for valence classification: 62.5 ± 15.8% (95% CI: 51.2–73.8%, *t*(9) = 2.37, *p* = 0.04) for GNB, 63.7 ± 13.1% (95% CI: 54.3–73.1%, *t*(9) = 3.16, *p* = 0.01) for SVM, and 68.8 ± 15.1% (95% CI: 58.0–79.6%, *t*(9) = 3.74, *p* < 0.01) for MLP. The SVM classifier utilizing right–left TMT achieved above-chance-level (>25%) accuracy in discrete emotion classification with an accuracy of 42.5 ± 16.0% (95% CI: 31.1–53.9%, *t*(9) = 3.28, *p* < 0.01). On the other hand, none of the classifiers utilizing right–left wrist skin temperature achieved accuracies higher than the chance level for either valence or discrete emotion classifications. Some of the observed accuracies of the wrist temperature model were significantly lower than chance level: 40.0 ± 13.5% (95% CI: 30.3–49.7%, *t*(9) = −2.34, *p* = 0.04) for valence classification by SVM; 18.8 ± 8.4% (95% CI: 12.8–24.8%, *t*(9) = −2.33, *p* = 0.04); and 16.2 ± 9.8% (95% CI: 9.2–23.2%, *t*(9) = −2.84, *p* = 0.02) for discrete emotion classification by GNB and SVM, respectively.

Figure 4 illustrates the comparison of the highest accuracy among the three classifiers for each classification task between models utilizing right–left TMT and right–left wrist skin temperature. Across all classification tasks, the highest accuracies for the right–left TMT model were higher than those of the right–left wrist skin temperature model.

## 6. Discussion

In this study, we measured the TMT and wrist skin temperature of participants as they experienced various emotions using two methods: autobiographical recall and scenario imagination. In both experiments, cluster-based analysis showed a difference in right–left TMT between positive and negative emotional conditions, and classifiers using the right–left TMT feature improved accuracy for valence and discrete emotion classification. In contrast, we did not find a difference in right–left wrist skin temperature under different emotional conditions, and classifiers based on right–left wrist skin temperature did not improve in classification accuracies.

The analysis of the temporal mean revealed a significant difference in right–left TMT between positive and negative emotions in the experiments, confirmed by cluster-based statistical testing. In both experiments, right–left TMT in negative emotional conditions was lower than that in positive emotional conditions in the clusters. One possible explanation for the observed difference is asymmetry in hemispheric activity under different emotional conditions. Previous studies on TMT during cognitive tasks suggested that increased hemispheric activity is associated with decreased TMT on the same side [18,19,20]. In normal conditions, brain temperature is warmer than body temperature, and it is considered that an increase in brain activity will increase blood flow from the cooler body to the brain. Researchers have assumed this would decrease the temperature of the tympanic membrane [21]. We can explain the observed difference by combining this assumption with the lateralized brain theory of emotion [26]. The lateralized brain hypothesis predicts that right lateralization in brain activity during negative emotional conditions leads to increased cerebral blood flow in the right side of the brain than the left. This may decrease right TMT than the left, which makes right–left TMT lower. The lateralized activation becomes vice versa during positive emotional conditions, and right–left TMT in negative emotion becomes lower than that in positive emotion under the assumptions. However, many other factors such as the outside environment or anatomical difference between both sides of the ears should also influence the result, and further investigation into the relationship between right–left TMT and emotion are necessary. In contrary, we did not find a significant cluster of the mean right–left wrist skin temperature among different emotional conditions. It may be reasonable to consider that lateralized brain activation was reflected in right–left TMT due to its proximity to the brain, but the effect of the lateralized activities did not reach temperature in peripheral areas such as wrists.

Importantly, we found that the change in right–left TMT was not consistent over time. The difference between positive and negative emotion was found in some periods of the stimuli, but not over the entire part. Therefore, if we had used the measurement in previous studies [21,22,23], which only measured TMT before and after the stimuli, we could not capture the differences in the middle of stimuli that were observed in the present study. While the block-wise assignment of the emotional labels limits further validation of the observed temporal dynamics, our findings suggest that the temporal variations are a promising feature of TMT that could provide further information about TMT-based emotional experience. Continuous TMT measurement has a higher power to detect the difference between the conditions, and it is expected to advance the research on the relationship between TMT and emotion.

The classification results show that right–left TMT can improve accuracy for predicting both valence and discrete emotions in different experiments. Multiple classifiers utilizing the same features of right–left TMT achieved higher accuracy than the chance level accuracy for discrete emotion classification and for valence emotion classification across experiments. It is notable that we only used right–left TMT to predict emotions, and we used the same features for inter-participant cross-validation in both experiments. The simplicity of our model is supportive of the generalizability of our findings that lateralization of TMT is useful for estimating emotion in various situations. On the other hand, the models utilizing right–left wrist temperature did not achieve higher accuracy than the chance level accuracy in either experiment, or the classification accuracies of the models were consistently lower than those of right–left TMT models. In Experiment 2, the accuracies of several classifiers were significantly lower than chance level. Since we employed leave-one-participant-out cross-validation for assessment, this result suggests a substantial difference in right–left wrist temperatures among participants that was not learned by the employed classifiers. Right–left TMT may provide unique information that right–left wrist temperature does not have, and the information is shown to be useful for predicting emotions.

The current study introduced TMT measurement by earphones, and its convenience was apparent during the experiments. Since participants just needed to wear earphones to start measuring TMT, the preparation time was remarkably shorter than that of EEG measurement, which requires calibration of all electrodes before the experiment. With the advantage of accessibility, future studies can measure TMT in various situations using earphone-type thermometers to find practical applications of emotion recognition in real life. For example, emotional experience in naturalistic environments such as music listing, video watching, or online conversation would be practical and valuable application. Lastly, TMT might be combined with other data sources, such as EEG and PPG, as well as accelerator data, which can also be measured by earphones [13,14,45]. A multimodal approach can capture diverse aspects of emotions [46,47,48], which could contribute to a robust estimation of emotion in real life. Advancements in sensor technology enable earphones to collect various data that will enhance the power to estimate the user’s emotional states.

Admittedly, there are some limitations to be noted. First, the current study was conducted with ten participants for the purpose of feasibility evaluation. Hereby, it is necessary to conduct further studies with larger sample sizes before moving toward practical applications. Second, although our model employing right–left TMT improved the classification accuracy of emotion recognition, the underlying mechanism between TMT lateralization and emotion remains largely unknown. Further investigation into the relationship between TMT lateralization and emotion is warranted. Third, we designed earphone-shaped thermometers for naturalistic applications; however, we did not collect behavioral and environmental data in the current study. Extended data collection in real-life scenarios, as in previous studies [49], would further advance the application of emotion recognition by TMT in wild environments. Finally, all our experiments were conducted with college students in a drama club, who are experienced in various emotion expressions, and the physiological signals induced in this sample may have different characters than those by general population [50]. Although this could benefit the present study on the development of new devices, future experiments with a broader population are needed to explore the generalizability of the present findings.

## 7. Conclusions

The current study developed earphone-type TMT thermometers and introduced right–left TMT as a novel metric for emotion recognition. Using two experiments with different emotion induction methods, we evaluated the effectiveness of this metric. Classifiers utilizing only right–left TMT achieved accuracies higher than chance levels in both valence classification and discrete emotion classification across experiments. Notably, improvement in classification accuracy was observed only in the model utilizing right–left TMT, and not with the model utilizing right–left wrist skin temperature. These findings suggest that right–left TMT can provide unique and useful information to predict emotions in different settings. The accessibility of TMT measurement by earphones highlights their significant potential for daily life applications in emotion recognition.

## Figures and Tables

**Figure 1 sensors-25-04411-f001:**
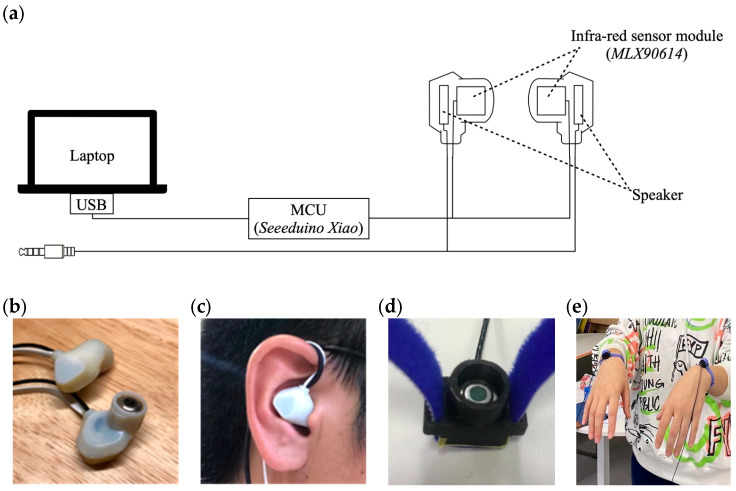
(**a**–**c**): The earphone-type TMT thermometers developed for the experiments. (**d**,**e**): The dorsal wrist skin temperature thermometer developed for the experiments.

**Figure 2 sensors-25-04411-f002:**
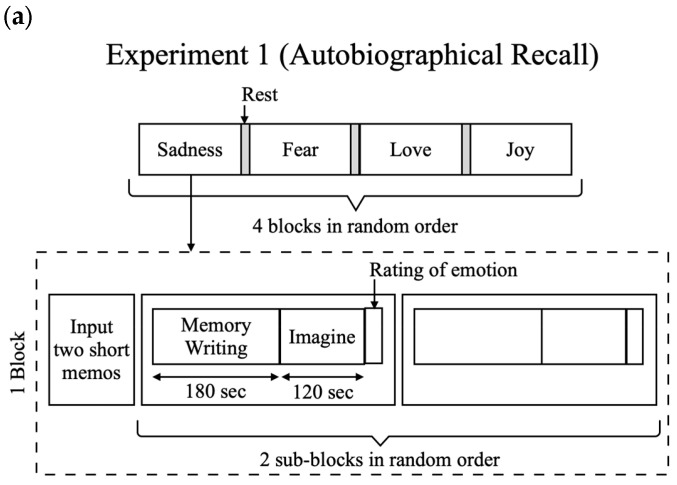
(**a**): Experimental procedures for autobiographical recall experiment (Experiment 1). (**b**): Experimental procedures for scenario imagination experiment (Experiment 2).

**Figure 3 sensors-25-04411-f003:**
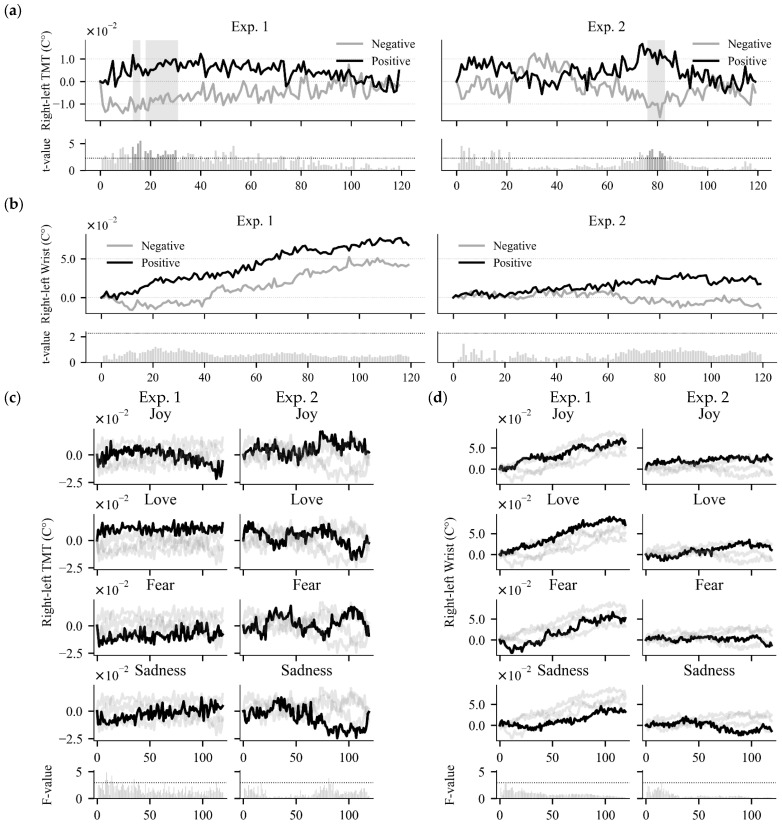
(**a**,**b**) The mean temperature—(**a**) right–left TMT and (**b**) right–left wrist skin temperature—grouped by positive or negative emotion tasks. The gray areas indicate temporal clusters where *p* < 0.05 in the non-parametric cluster-based testing. The bottom bar plots display the absolute *t*-values from paired *t*-tests at each time point within the tasks, with horizontal dotted lines indicating the threshold (*t*(9) = 2.26). (**c**,**d**) The mean temperature—(**c**) right–left TMT and (**d**) right–left wrist skin temperature—grouped by targeted discrete emotions. The bold black line represents the mean temperature for the target emotion, and the gray lines represent those for others. The bottom bar plots illustrate *F*-values from the repeated measures ANOVA at each time point within the tasks, with horizontal dotted lines indicating the threshold (*F*(3,27) = 2.96).

**Figure 4 sensors-25-04411-f004:**
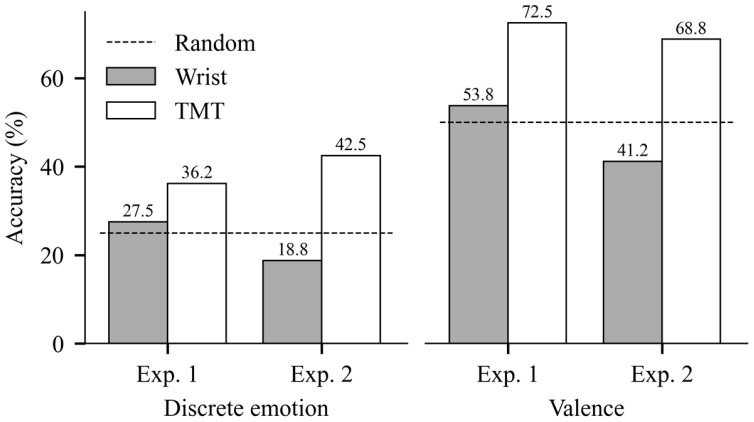
The highest accuracies at each classification task by wrist temperature and TMT model. The dashed horizontal lines show the random chance level. The **left** plot is from discrete emotion classification and the **right** plot is from valence classification.

**Table 1 sensors-25-04411-t001:** Descriptive statistics of emotion ratings from Experiment 1 and 2.

		Emotion Ratings							
Exp.	Target Emotion	Joy	Tenderness	Inspire	Amusement	Anger	Disgust	Fear	Sadness
Exp. 1	Positive	**6.6 ± 1.8**	6.0 ± 2.5	5.0 ± 2.9	3.2 ± 2.4	1.8 ± 1.6	1.4 ± 1.2	1.6 ± 1.2	3.3 ± 2.4
	Joy	**7.3 ± 0.9**	5.4 ± 2.6	5.5 ± 2.6	4.2 ± 2.5	1.3 ± 0.6	1.2 ± 0.5	1.6 ± 1.3	2.3 ± 1.6
	Love	5.8 ± 2.2	**6.6 ± 2.4**	4.4 ± 3.1	2.2 ± 1.8	2.3 ± 2.1	1.6 ± 1.6	1.7 ± 1.2	4.4 ± 2.7
	Negative	1.5 ± 0.7	1.8 ± 1.5	1.8 ± 1.6	1.3 ± 0.8	4.2 ± 2.6	4.3 ± 2.9	5.7 ± 2.7	**6.2 ± 2.6**
	Fear	1.4 ± 0.7	1.2 ± 0.5	1.8 ± 1.7	1.5 ± 0.9	3.8 ± 2.5	4.3 ± 3.0	**7.6 ± 1.0**	5.1 ± 3.1
	Sadness	1.5 ± 0.8	2.4 ± 1.9	1.8 ± 1.6	1.2 ± 0.5	4.5 ± 2.6	4.4 ± 2.9	3.8 ± 2.5	**7.4 ± 1.0**
Exp. 2	Positive	**7.0 ± 1.2**	6.8 ± 1.9	5.7 ± 2.7	3.0 ± 2.0	1.2 ± 0.4	1.1 ± 0.2	1.1 ± 0.4	2.1 ± 1.7
	Joy	**7.3 ± 0.9**	6.2 ± 2.3	6.4 ± 2.4	3.1 ± 2.2	1.2 ± 0.4	1.1 ± 0.3	1.2 ± 0.4	1.3 ± 0.7
	Love	6.6 ± 1.3	**7.5 ± 1.3**	5.0 ± 2.9	2.9 ± 2.0	1.2 ± 0.4	1.1 ± 0.1	1.1 ± 0.3	2.9 ± 2.0
	Negative	1.2 ± 0.6	1.6 ± 1.3	1.4 ± 1.1	1.1 ± 0.2	3.3 ± 2.6	3.4 ± 2.6	6.0 ± 3.0	**6.4 ± 2.4**
	Fear	1.2 ± 0.6	1.1 ± 0.3	1.5 ± 1.4	1.1 ± 0.3	3.4 ± 2.6	4.5 ± 2.6	**8.1 ± 0.9**	4.9 ± 2.4
	Sadness	1.2 ± 0.7	2.1 ± 1.7	1.3 ± 0.7	1.1 ± 0.2	3.1 ± 2.7	2.4 ± 2.2	3.9 ± 2.9	**8.0 ± 1.0**

Each rating was scaled from 1 (weakest) to 9 (strongest). The table presents the mean and *SD* of emotion ratings for discrete emotion (joy, love, fear, or sadness) and valence (positive emotion or negative emotion). In each experiment, **bold** numbers indicate the highest mean emotion rating in each target emotion within both positive/negative emotion and discrete emotion.

**Table 2 sensors-25-04411-t002:** Classification accuracy (%) from Experiment 1.

Class	Data	Chance	GNB	SVM	MLP
Valence	TMT	50.0	**63.7 ± 13.1**	**67.5 ± 13.9**	**72.5 ± 16.6**
	Wrist		53.8 ± 21.0	50.0 ± 20.9	43.8 ± 17.0
Discrete	TMT	25.0	**36.2 ± 8.8**	**35.0 ± 12.2**	35.0 ± 13.5
	Wrist		27.5 ± 10.9	21.2 ± 8.0	27.5 ± 9.4

The **bold** is higher than the chance level (*p* < 0.05, uncorrected).

**Table 3 sensors-25-04411-t003:** Classification accuracy (%) from Experiment 2.

Class	Data	Chance	GNB	SVM	MLP
Valence	TMT	50.0	**62.5 ± 15.8**	**63.7 ± 13.1**	**68.8 ± 15.1**
	Wrist		41.2 ± 14.8	40.0 ± 13.5	38.8 ± 20.5
Discrete	TMT	25.0	32.5 ± 15.0	**42.5 ± 16.0**	36.2 ± 15.3
	Wrist		18.8 ± 8.4	16.2 ± 9.8	18.8 ± 12.8

The **bold** is higher than the chance level (*p* < 0.05, uncorrected).

## Data Availability

Data and code of this work will be available from the corresponding author upon reasonable request.

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
