# Peer review of "Utilizing Tympanic Membrane Temperature for Earphone-Based Emotion Recognition"

_sensors, 2025, doi:10.3390/s25144411_

Round 1

Reviewer 1 Report

Comments and Suggestions for Authors

The citations are not recent, but this does not affect the relevance of the work. Given the nature of the research, it can be considered irrelevant.

Author Response

First of all, we would like to thank you, the editor Ms. Sylvie Yao and the four reviewers for the time and effort spent in reviewing our manuscript. Those comments are valuable and helpful for improving our paper. Please find a point-by-point reply below, followed by the revised manuscript with all the changes marked in red.

Comments 1: The citations are not recent, but this does not affect the relevance of the work. Given the nature of the research, it can be considered irrelevant.

Response 1: We have included the following recent publications in the affective sensing field to better position the present study.

2. Wang, Y.; Song, W.; Tao, W.; Liotta, A.; Yang, D.; Li, X.; Gao, S.; Sun, Y.; Ge, W.; Zhang, W.; et al. A Systematic Review on Affective Computing: Emotion Models, Databases, and Recent Advances. Inf. Fusion 2022, 83–84, 19–52.    (page 2, line 36)

46. Guo, R.; Guo, H.; Wang, L.; Chen, M.; Yang, D.; Li, B. Development and Application of Emotion Recognition Technology - a Systematic Literature Review. BMC Psychol. 2024, 12, 95 (page 14, line 467)

47. Li, F.; Zhang, D. Transformer-Driven Affective State Recognition from Wearable Physiological Data in Everyday Contexts. Sensors (Basel) 2025, 25, doi:10.3390/s2503076 (page 14, line 467)

48. Yin, Y.; Kong, W.; Tang, J.; Li, J.; Babiloni, F. PSPN: Pseudo-Siamese Pyramid Network for Multimodal Emotion Analysis. Cogn. Neurodyn. 2024, 18, 2883–2896. (page 14, line 467)

50. Shui, X.; Zhang, M.; Li, Z.; Hu, X.; Wang, F.; Zhang, D. A Dataset of Daily Ambulatory Psychological and Physiological Recording for Emotion Research. Scientific Data 2021, 8, 1–12 (page 15, line 483)

Reviewer 2 Report

Comments and Suggestions for Authors

The article is interesting, and it contributes to an area where further research is required. However, I recommend several minor improvements.

I recommend to the authors to describe in more detail the earphone-based emotion recognition equipment i.e. the utilized sensor and its characteristics (sensitivity, resolution e.t.c)

Furthermore, what is the cost of the proposed equipment.

Finally, the sample size is quite small, thus further evaluation activities must be carried out with larger samples.

Author Response

First of all, we would like to thank you, the editor Ms. Sylvie Yao and the four reviewers for the time and effort spent in reviewing our manuscript. Those comments are valuable and helpful for improving our paper. Please find a point-by-point reply below, followed by the revised manuscript with all the changes marked in red.

Comments 1: I recommend to the authors to describe in more detail the earphone-based emotion recognition equipment i.e. the utilized sensor and its characteristics (sensitivity, resolution e.t.c)

Response 1: The following sentence was updated to include a more detailed description of devices as advised (page 4, line 162–169):

We used an infrared sensor (MLX90614ESF-DCA, Melexis) with medical accuracies for 36 to 38 °C and a resolution of 0.02°C, which was designed to measure human body temperature. The sensor was factory-calibrated, and the temperature came with digital output via SMBus. The data was sampled at 20 Hz using a microcontroller (SEEEDUINO XIAO, Seeed Technology) equipped with a low-power ARM Cortex-M0+ processor. The circuit used an I2C communication setup where a microcontroller connected to a computer reads data from two infrared temperature sensors.

Comments 2: Furthermore, what is the cost of the proposed equipment.

Response 2: The following sentence explaining the device cost was included in Material and Methods as suggested (page 4-5, line 173–175):

The material cost for one set of earphones or wristbands was approximately USD 55 (infrared sensors: USD 44; microcontroller: USD 5; other components: USD 6).   

Comments 3: Finally, the sample size is quite small, thus further evaluation activities must be carried out with larger samples.

Response 3: The following sentence was included in the discussion part as advised to clarify limitation of our study (page 14, line 470–473):

Admittedly, there are some limitations to be noted. First, the present study was conducted with ten participants for the purpose of feasibility evaluation. Hereby, it is necessary to conduct further studies with larger sample sizes before moving towards practical applications. 

Reviewer 3 Report

Comments and Suggestions for Authors

This manuscript describes work to compare in ear EEG and lateralization of signals to generate correlates of emotional state. They add in tympanic membrane temperature as a novel signal to help. They show some modest improvement but more importantly, they highlight the specific information gain from TMT compared to other common wearable sensor locations for temperature measurement. This is important for those seeking to know which temperature sensor arrangement might provide optimal input for emotion classification tasks.

The introduction and related work sections provide a nice argument for the need of physiological data in emotional assessment. They follow with a well structured argument for the unique potential of TMT. Very clearly articulated. The discussion is likewise thoughtful and clear.

I don’t know for this journal, but most request specific IRB approval numbers when describing approvals.

Specific comments:

  1. I would be interested to hear why blocks are the length they are. Emotions are hard to quantify or even define, and as such can exist at different timescales. For example, one maybe acutely happy after hearing a joke, despite being in the middle of a week long depression. I would be curious to see some mention of how the experimenters decided on the timescales they chose.
  2. In Ln 282-283 “two tailed p values were calculated” how? What test seems to be missing.
  3. It’s a small point, but being different from chance is the same in both directions from an information point of view. It would still be worth assessing whether wrist temperature is different from chance, even if worse (in which case, flip the sign in your classifier). The performance is still worse clearly (i.e. Exp 2 Valence) but might be nonetheless non-random.
  4. I’m surprised there is not mention of device calibration. Since the authors state they made their own devices for measurement, I would like control specs to know the devices measure what they’re supposed to. Hopefully this is an easy addition, but I believe it is necessary to ensure correct interpretation.
  5. I’m also surprised not to see mention of the subjects being actors in the discussion. Is it important someone is practiced at focusing on an emotion for detection to work? What might this say about stability and or the needs of improvement in less emotionally-aware and or more authentically/naturally occurring emotion classification?

Author Response

First of all, we would like to thank you, the editor Ms. Sylvie Yao and the four reviewers for the time and effort spent in reviewing our manuscript. Those comments are valuable and helpful for improving our paper. Please find a point-by-point reply below, followed by the revised manuscript with all the changes marked in red.

Comments 1: I don’t know for this journal, but most request specific IRB approval numbers when describing approvals.

Response 1: IRB approval number was included in the 3.2 Overview of Experiments as advised (page 5, line 192):

Experiment 1 was conducted alongside Experiment 2, and ethical approvals for both experiments were obtained from the local ethical committee at the Department of Psychological and Cognitive Sciences, Tsinghua University (IRB Approval 2023 No. 12).

Comments 2: I would be interested to hear why blocks are the length they are. Emotions are hard to quantify or even define, and as such can exist at different timescales. For example, one maybe acutely happy after hearing a joke, despite being in the middle of a week long depression. I would be curious to see some mention of how the experimenters decided on the timescales they chose.

Response 2: Admittedly, emotions can fluctuate at finer or coarser scales depending on the context. However, as the key research question is about the feasibility of a new measurement approach, here we took a conservative approach that followed the time length in classical emotion research paradigms in previous physiological studies, which was now explained in 3. Materials and Methods (page 6, line 211–214):

The time length of each task was decided referencing a previous study that measured TMT before and after five-minutes emotion induction session [23], which was also comparable to the minute-level time scales commonly used in other physiological studies on emotion [38–41].

In addition, the following sentences were included to discuss temporal fluctuation of TMT for emotion representation in 6. Discussion (page 14, line 434–438):

While the block-wise assignment of the emotional labels limits further validation of the observed temporal dynamics, our findings suggest that the temporal variations are a promising feature of TMT that could provide further information about TMT-based emotional experience.

Comments 3: In Ln 282-283 “two tailed p values were calculated” how? What test seems to be missing.

Response 3: The following sentence was updated to clarify the test method used to calculate the p-value (page 8, line 289-293):

The observed cluster-level statistics were then compared to the null distribution, and (two-tailed) p-values were calculated as the proportion of permutations in the null distribution where the absolute value of the permutation's cluster-level test statistic was larger than or equal to the absolute value of the observed cluster-level test statistics.

Comments 4: It’s a small point, but being different from chance is the same in both directions from an information point of view. It would still be worth assessing whether wrist temperature is different from chance, even if worse (in which case, flip the sign in your classifier). The performance is still worse clearly (i.e. Exp 2 Valence) but might be nonetheless non-random.

Response 4: The reported accuracies were cross-validated. Hereby, accuracy lower than chance level suggests that the learned data patterns by the classifier could not be informative about the true underlying class distinctions, possibly due to mislabeled data, inverted learning signals, or spurious correlations that misled the model.
The following sentences are updated to report the significance (page 12, line 382–386):

Some of the observed accuracies of the wrist temperature model were significantly lower than chance level: 40.0 ±â€¯13.5% (95% CI: 30.3–49.7%, t(9) = -2.34, p = 0.04) for valence classification by SVM; 18.8 ±â€¯8.4% (95% CI: 12.8–24.8%, t(9) = -2.33, p = 0.04); and 16.2 ±â€¯9.8% (95% CI: 9.2–23.2%, t(9) = -2.84, p = 0.02) for discrete emotion classification by GNB and SVM, respectively.

Also, the following sentence was included to explain the results in 6. Discussion (page 14, line 451–454):

In Experiment 2, the accuracies of several classifiers were significantly lower than chance level. Since we employed leave-one-participant cross-validation for assessment, this result suggests a substantial difference in right-left wrist temperatures among participants that was not learned by the employed classifiers.

Comments 5: I’m surprised there is not mention of device calibration. Since the authors state they made their own devices for measurement, I would like control specs to know the devices measure what they’re supposed to. Hopefully this is an easy addition, but I believe it is necessary to ensure correct interpretation.

Response 5: The sensor used in our study was calibrated in the supplier’s factory. The following sentence was updated to clarify about calibration for the equipment (page 4, line 162–165):

We used an infrared sensor (MLX90614ESF-DCA, Melexis) with medical accuracies for 36 to 38 °C and a resolution of 0.02°C, which was designed to measure human body temperature. The sensor was factory-calibrated, and the temperature came with digital output via SMBus.

Comments 6: I’m also surprised not to see mention of the subjects being actors in the discussion. Is it important someone is practiced at focusing on an emotion for detection to work? What might this say about stability and or the needs of improvement in less emotionally-aware and or more authentically/naturally occurring emotion classification?

Response 6: The following sentence was included in 6. Discussion to clarify limitation of our study (page 14-15, line 480–485):

Finally, all our experiments were conducted with college students in a drama club, who are experienced in various emotion expressions, and the physiological signals induced in this sample may have different characters than those by general population [50]. Although this could benefit the present study on the development of new devices, future experiments with a broader population are needed to explore the generalizability of the present findings.

Reviewer 4 Report

Comments and Suggestions for Authors

The paper is innovative and interesting. The authors should describe the process and results in a more systematic manner. Comments that may be considered by the authors: 

  • Identifying that TMT differences between emotions are time-specific rather than uniform is promising. Pls elaborate on the temporal resolution of the measurements including the identification of the windows . Were statistical comparisons performed across time segments, or is the conclusion based on visual inspection?
  • The authors should provide more detail from describing both process and results, including confidence intervals, effect sizes, and statistical significance for the observed accuracies.
  • Drama students have been selected. We propose to validate the results (in terms of generalizability) across individuals.
  • similarly the authors are encouraged to investigate the behaviour in uncontrolled naturalistic environments (e.g., walking and or talking) 
  • Investigate multimodal integration  with other physiological signals such as EEG. 

Author Response

First of all, we would like to thank you, the editor Ms. Sylvie Yao and the four reviewers for the time and effort spent in reviewing our manuscript. Those comments are valuable and helpful for improving our paper. Please find a point-by-point reply below, followed by the revised manuscript with all the changes marked in red.

Comments 1: Identifying that TMT differences between emotions are time-specific rather than uniform is promising. Pls elaborate on the temporal resolution of the measurements including the identification of the windows.

Response 1: The recording resolution (20Hz) is now reported (page 4, line 165). The following sentences were included to discuss about time length of TMT in 6. Discussion (page 14, line 434–438):

While the block-wise assignment of the emotional labels limits further validation of the observed temporal dynamics, our findings suggest that the temporal variations are a promising feature of TMT that could provide further information about TMT-based emotional experience.

Comments 2: Were statistical comparisons performed across time segments, or is the conclusion based on visual inspection?

Response 2: The following sentence was revised to clarify the statistical method used for the analysis (page 8, line 289-293):

The observed cluster-level statistics were then compared to the null distribution, and (two-tailed) p-values were calculated as the proportion of permutations in the null distribution where the absolute value of the permutation's cluster-level test statistic was larger than or equal to the absolute value of the observed cluster-level test statistics.

Comments 3: The authors should provide more detail from describing both process and results, including confidence intervals, effect sizes, and statistical significance for the observed accuracies.
Response 3: The following sentences were updated to mention confidence intervals as suggested (page 12, line 362–369, 374-380):

The classification model using right-left TMT achieved accuracies higher than chance level (>50%) in positive versus negative (valence) classification for all classifiers: 63.7±13.1% (95% CI: 54.3-73.1%, t(9) = 3.16, p = 0.01) for GNB, 67.5±13.9% (95% CI: 57.6-77.4%, t(9) = 3.77, p < 0.01) for SVM, and 72.5±16.6% (95% CI: 60.6-84.4%, t(9) = 4.07, p < 0.01) for MLP. The right-left TMT model also achieved above-chance-level (>25%) accuracy in discrete emotion classification with accuracies of 36.2±8.8% (95% CI: 29.9-42.5%, t(9) = 3.86, p < 0.01) for GNB and 35.0±12.2% (95% CI: 26.3-43.7%, t(9) = 2.45, p = 0.03) for SVM.

All classifiers utilizing right-left TMT achieved accuracies higher than the chance level (>50%) for valence classification: 62.5±15.8% (95% CI: 51.2-73.8%, t(9) = 2.37, p = 0.04) for GNB, 63.7±13.1% (95% CI: 54.3-73.1%, t(9) = 3.16, p = 0.01) for SVM, and 68.8±15.1% (95% CI: 58.0-79.6%, t(9) = 3.74, p < 0.01) for MLP. The SVM classifier utilizing right-left TMT achieved above-chance-level (>25%) accuracy in discrete emotion classification with an accuracy of 42.5±16.0% (95% CI: 31.1-53.9%, t(9) = 3.28, p < 0.01).

Comments 4: Drama students have been selected. We propose to validate the results (in terms of generalizability) across individuals.

Response 4: The following sentence was revised to clarify the limitation of our study in the Discussion part (page 14-15, line 480–485):

Finally, all our experiments were conducted with college students in a drama club, who are experienced in various emotion expressions, and the physiological signals induced in this sample may have different characters than those by general population [50]. Although this could benefit the present study on the development of new devices, future experiments with a broader population are needed to explore the generalizability of the present findings.

Comments 5: similarly the authors are encouraged to investigate the behaviour in uncontrolled naturalistic environments (e.g., walking and or talking)

Response 5:  The following sentence was included as limitation of our study and to illustrate the potential of naturalistic environment in Discussion part (page 14, line 463–464):

For example, emotional experience in naturalistic environments such as music listing video watching, or online conversation would be practical and valuable application.

In addition, the following sentences were included to describe the limitations of our studies. (page 14, line 476–480):

Third, we designed earphone-shaped thermometers for naturalistic applications, however, we did not collect behavioral and environmental data in the current study. Extended data collection in real-life scenarios, as in previous studies [38], would further advance the application of emotion recognition by TMT in wild environments. 

Comments 6: Investigate multimodal integration with other physiological signals such as EEG.

Response 6: The following sentence was included to illustrate the potential of multimodal approach in the Discussion part (page 14, line 465–469):

Lastly, TMT might be combined with other data sources, such as EEG, PPG as well as accelerator data, which can also be measured by earphones [13,14,45]. A multimodal approach can capture diverse aspects of emotions [46–48], that could contribute to robust estimation of emotion in real life. Advancements in sensor technology enable earphones to collect various data that will enhance the power to estimate the user's emotional states.